# The Effect of Music on Livestock: Cattle, Poultry and Pigs

**DOI:** 10.3390/ani11123572

**Published:** 2021-12-16

**Authors:** Patrycja Ciborowska, Monika Michalczuk, Damian Bień

**Affiliations:** Department of Animal Breeding, Institute of Animal Sciences, Warsaw University of Life Sciences, Ciszewskiego 8, 02-786 Warsaw, Poland; patrycja.ciborowska21@gmail.com

**Keywords:** music therapy, music, music genre, sound waves, livestock production, welfare

## Abstract

**Simple Summary:**

In times of intensified livestock production, the search for methods that reduce stress, which has an adverse impact on the health and welfare of their animals, has become a challenge for breeders and producers. Therefore, the possibility of using various musical genres to alleviate stress in chickens, cattle or pigs was considered. It has turned out that choosing a musical item is extremely important, as it can positively affect the health and production performance of animals by increasing the feeling of relaxation. The time of exposure to sounds and their intensity are important as well, and some authors propose to also pay attention to the frequency of sound waves. Music therapy, which was previously more widely deployed among humans, is increasingly used for farm animals as an element of enriching their living environment. Current research shows the importance of sound waves’ influence in animal production. Proper selection of the music genre, music intensity and tempo can reduce the adverse effects of noise and, thus, reduce the level of stress. It should be remembered, however, that silence is equally important and necessary for the welfare of animals. The paper presents literature findings regarding the influence of music on cattle, poultry and pigs.

**Abstract:**

The welfare of animals, especially those kept in intensive production systems, is a priority for modern agriculture. This stems from the desire to keep animals healthy, to obtain a good-quality final product, and to meet the demands of today’s consumers, who have been increasingly persuaded to buy organic products. As a result, new sound-based methods have been pursued to reduce external stress in livestock. Music therapy has been known for thousands of years, and sounds were believed to improve both body and spirit. Today, they are mostly used to distract patients from their pain, as well as to treat depression and cardiovascular disorders. However, recent studies have suggested that appropriately selected music can confer some health benefits, e.g., by increasing the level and activity of natural killer cells. For use in livestock, the choice of genre, the loudness of the music and the tempo are all important factors. Some music tracks promote relaxation (thus improving yields), while others have the opposite effect. However, there is no doubt that enriching the animals’ environment with music improves their welfare and may also convince consumers to buy products from intensively farmed animals. The present paper explores the effects of music on livestock (cattle, poultry and pigs) on the basis of the available literature.

## 1. Introduction

Animal welfare has become a widely discussed issue. In an era of intensive animal farming, modern agriculture seeks to improve the welfare of reared animals while ensuring good quality of the resultant animal products. There is also increased focus on the satisfaction of consumers, who increasingly tend to follow consumer trends (e.g., “mindful eating” or “respect for animals”) and have become more discerning about the perceived quality and origin of their food [1,2]. Just as often, the choice of animal products is driven by ethical and environmental considerations, which is best exemplified by the increasing popularity of organic products [3]. However, there is a large segment of price-conscious consumers who tend to choose cage-laid eggs or conventional milk rather than buying organic due to limited household income. To meet the demands of today’s consumers, new methods of improving animal welfare have been sought in order to convince the segment of the public that questions the use of intensive farming systems [4].

The concept of well-being refers to a state where an individual is able to cope with environmental pressures [5]. Under optimal rearing conditions, animals are expected to maintain an adequate level of physical and mental health. Breeders and producers achieve this through balanced feeding, constant access to fresh water, use of enclosures, interaction with other animals, and prevention/treatment in accordance with the principles of Five Freedoms [6]. New methods are being sought to reduce the impact of stress triggers, which cause multiple problems, including reduced yields. Methods used to improve well-being include various toys for pigs [7], mechanical brushes for cattle [8], functional feed additives [9,10], or limited contact with humans [11]. Recently, there has been growing interest in the subject of sound waves as a way to alleviate the negative effects of stress in animals kept in intensive production systems. The first trials on the effect of music on animals were reported in the 20th century [12,13,14]. Cattle is the most popular subject for such research, but successful trials have also been conducted with horses, and even carps and trouts [15,16,17,18]. In addition to musical genres, there have also been publications focusing on specific soundwave frequencies as determinants of animal health [19,20].

Though there have been many definitions of music, it is usually defined as “an art of sound in time that expresses ideas and emotions in significant forms through the elements of rhythm, melody, harmony, and color” or “the tones or sounds employed, occurring in single line (melody) or multiple lines (harmony), and sounded or to be sounded by one or more voices or instruments, or both” [21]. It can also be certainly defined as a combination of various elements, such as rhythm, tone, frequency, loudness, and lyrics [22]. For thousands of years, music was viewed in the same light as medicine. In Ancient times, the Greek god Apollon was both the patron of music, and of healing/medicine. Thus, prior to the Medieval period, much significance was attributed to musical scales (ratios between sound intervals, which are used as the basis to compose and perform music representative of the given age/culture) [23]—such as the Doric or Phrygian scales, which were thought by ancient philosophers (such as Boethius) to elicit specific behaviors—such as inebriation—depending on the scale used. Sisthaltic, disthatic and hesicastic compositions were seen as depressing, “uplifting”, and soothing, respectively [24]. It was claimed that music could help not only the soul, but the body as well, by influencing the physiology of the human body [25]. Today, it is used to distract patients from their pain, as well as to treat depression and cardiovascular disorders [26,27,28]. However, it has been shown that appropriately chosen sounds can confer health benefits, e.g., by increasing the level and activity of natural killer (NK) cells—one of the immune system’s lines of defense [29]. Data from various studies on humans have been used to develop methods of using sound waves on animals.

## 2. The Problem of Stress in Livestock

There is no doubt that the intensification of livestock farming has exasperated stress among animal species grown for various products. The father of the concept of stress—Hans Hugo Selye—has said that stress is a natural part of everyday life, and should be treated as a normal response of the human body (with a specific pattern) to a particular stressor [30]. Walter B. Cannon further expanded this definition to encompass disruption of homeostasis and the body’s attempt to revert to optimal conditions through a series of physiological reactions [31]. Crucially, almost every physiological process is predisposed towards maintaining homeostasis, so the term “stress” should be restricted to unexpected situations that involve loss of control [32]. The popular concept of stress refers to the exposure to unpleasant conditions that ultimately produces deleterious effects—poor health and discomfort in general, and reduced yields and product quality in livestock specifically [33,34]. The term “distress” can also be distinguished, tentatively defined as “state in which an animal cannot escape from or adapt to the internal or external stressors or conditions it experiences, resulting in negative effects on its well-being” [35]. The concepts of “distress” and “stress” are similar to each other, which results from the authors’ frequent blurring of the boundary between them. However, the factors differentiating these two terms include, among others, the duration of the response and the intensity of the stressor [36]. Distress manifests itself in physiological and biochemical changes in the organism of animals, which in turn follow both chronic and acute stress, which ultimately leads to the organism being overloaded [37]. This means that stress can turn into distress, which puts a greater burden on the organism, which begins to use the available reserve resources to return to homeostasis [38]. An additional feature of distress is that it is difficult to diagnose solely through behavioral observations, e.g., subclinical pathological lesions (e.g., hypertension) that are not visible from the outside [36].

The stress response is mediated by two key factors—the sympathetic nervous system (SNS) and the hypothalamic–pituitary–adrenal axis (HPA). When suddenly exposed to danger, the sympathetic nervous system activates, inducing a fight or flight response in the body. The reaction starts in the adrenal glands, which are stimulated to release adrenaline and noradrenaline (catecholamine) (Figure 1), resulting in increased heart rate, pupil dilation and increased blood pressure. The body is prompted to use up its energy stores in muscles and the liver, increasing blood glucose by way of glycogenolysis [39].

If the stressogenic factor does not subside within about 15 min, the HPA is activated. This response begins in the hypothalamus, which secretes corticotropin-releasing hormone (CRH). This, in turn, stimulates the release of adrenocorticotropic hormone (ACTH) in the pituitary gland (Figure 2). ACTH is responsible for inducing the adrenal cortex to secrete glucocorticoids-hormones that play an important role in metabolic processes. The primary stress hormone in mammals is cortisol, followed by aldosterone and cortisone, whereas corticosterone is the dominant stress hormone in birds [39,40,41]. Acute spikes in the levels of these hormones induce lipolysis and proteolysis, and can even raise blood glucose levels [39]. During prolonged bouts of stress, the high blood levels of glucocorticoids lead to disorders that reduce organ weight (e.g., lymph nodes, spleen). As a result, the animal suffers from a weakened immune system and becomes more susceptible to pathogens [42].

The most common stress triggers affecting livestock are:Unbalanced portions of feed;Limited access to water;Rearing conditions (e.g., overcrowding);Separation/weaning of the young;Substandard zoohygienic conditions (e.g., humidity, temperature, noise);Transportation;Disease;Man–animal interaction.

Though livestock species have undergone some degree of selection to better tolerate the rearing/farming environment, each species still has to deal with a number of stress triggers [42,43,44]. Apart from behavioral problems, this can also cause excessive mortality and meat defects in pigs and poultry, e.g., PSE (pale, soft, exudative) or DFD (dark, firm, dry) meat [42,45,46]. In cattle, stress often results in reproductive disorders, and can even reduce milk yield/milkability; this may be related to inadequate care of the cow during milking by the worker as a result of which it feels fear [47,48].

There is no doubt that stress can negatively impact yields, and the resultant deleterious effects on animal health drive up costs [49]. Bearing this in mind, research is underway to reduce stress and associated impacts by improving animal well-being, e.g., by selecting specimens, optimizing rearing conditions, or even using feed additives [50,51].

## 3. The Use of Music

### 3.1. Music and Silence

Although environment enrichment with music can be an effective tool for stress reduction in livestock, periods of no auditory stimulation should also be provided [52]. Interspersing slow-paced and fast-paced music with approx. 2 min pauses (without any auditory stimulation) may be an effective way to treat cardiovascular disease. Music has been shown to improve breathing control in patients, whereas the pauses decreased blood pressure, heart rate and respiration rate. Furthermore, the subjects exhibited deeper relaxation during pauses after auditory stimulation, rather than the silent period preceding the music [53]. It has also been shown that Depth Relaxation Music Therapy (DRMT) in tandem with Hypnomusictherapy (HMT) and the quiet of the natural environment encouraged much deeper post-session relaxation among the subjects [54]. Sutton (2005) points out that silence can be a way of promoting engagement in patients under music therapy. Silence is as important to a song as sound—the pauses, known as “rests”, serve to punctuate the musical piece, allowing the musician and the listeners to take a moment to breathe and relax [55]. Kemp (2019) has found that some genres of music (such as country, classical music, lullaby) had a beneficial effect on bovine welfare, resulting in lower heart rates (HR) and respiration rates. Nevertheless, the highest milk yields were recorded for the control group, which was not exposed to any music [15]. In turn, Crouch et al. (2019) noted that auditory stimuli lessened to frequency of irregular behaviors in cattle (such as tongue rolling or vocalizations) and even promoted social interaction within the herd. Notably, however, lack of auditory stimulation proved to be just as important, prompting deeper rest among the animals, as well as intensified rumination, which seems to signify deeper relaxation and higher productivity [52]. A study by Ekachat and Vajrabukka (1994) showed that pigs exposed to light music (slow rhythm music) performed similarly to those kept in silence (no-music control). Final liveweight did not differ significantly from the no light music control [56]. There are studies that have used relaxing music [57]. While researchers have been unable to find a definition of this type of music, they have analyzed selected compositions defined as the relaxation genre from the repertoire of Enya (“Only Time”), Vangelis (“Conquest of paradise”) and Yanni (“Prelude and Nostalgia”)—the works of these composers were used in the study by Khalfa et al. (2003) (the article does not mention the titles of the compositions used; sample items were selected for the analysis). After an auditory analysis, it was found that relaxing music merges the warmly sounding instruments such as: flute, classical guitar, violin. It is played at a tempo of approx. 60–65 bpm (beats per minute), and the melody line is performed with the use of legato—articulation with which the notes are played smoothly, which creates the impression of a “melody wave” with the use of low rather than high notes. In the mentioned research work, Khalfa et al. (2003) showed that people without musical auditory stimulation had a higher salivary cortisol (stress hormone) level when exposed to a psychological stressor compared with subjects treated with relaxing music (from the repertoire of Enya, Vangelis, and Yanni) (Figure 3). In the music-treated group, salivary cortisol levels abated much quicker [57]. These findings substantiate the positive effects of relaxing music, while also showing that the benefits of silence are more pronounced when used intermittently with auditory stimuli (music). Therefore, it should be remembered that it is good practice to intersperse periods of music with breaks, both in humans, as noted by Bernardi et al. (2006), and in livestock, as mentioned by Crouch et al. (2019). These authors report that serene, slow-tempo music produces a relaxing effect, which is even more evident during breaks between auditory simulation [52,53]. It should be mentioned here that maintaining silence at the place where animals are kept and in the vicinity is also important for their welfare. This is possible by reducing the external noise of farm tractors, self-propelled machines and other devices, including garden devices used at farms, but also elements that may, for example, hit the building wall on windy days, or an unlubricated door to a livestock building, which is opened several times a day by workers and thereby can stress the animals. The analysis of year-round noise exposure, depending on the production profile, showed that by far the highest exposure to noise occurs at farms with crop production (90.3–91.4 dB), but also at those focused on livestock production [58]. Cattle, especially dairy cows, are animals that require silence during rearing. Their prolonged exposure to high-intensity noise may adversely affect the quantity and quality of milk produced, which then translates into economic losses on the part of the producer [59].

### 3.2. The Influence of Sound WAVES on Animals

The application of music therapy in humans has shown that well-selected sounds improve health and can even be regarded as a non-pharmacological treatment for various conditions [60]. Apart from increasing focus, acoustic waves of some musical pieces may alleviate pain, change heart rate (HR, HRV), reduce anxiety, reduce stress hormone (e.g., cortisol) production, and even significantly improve NK cell levels and activity [22,61,62]. Sounds used for music therapy should be carefully selected so as not to cause stress for the animal. Though music can certainly be used as environmental enrichment for many species, the genre is a significant factor as well. According to research, the most health-promoting genres include classical music (such as Mozart, Bach), relaxing music, and meditation music, whereas listeners of techno and heavy metal are at risk of higher stress, or even heart arrhythmia [63]. Snowdon et al. (2015) drew from these findings to compose music for domestic cats. The authors’ hypothesis held that the music should harness the frequencies and tempos that naturally occur in cat communication. This assumption proved correct—the species-specific music spurred cats to exhibit increased activity such as purring and rubbing against the source of sound. The cats were less responsive to human music [64]. This study shows that, much like humans, animals can psychologically interpret musical pieces by showing interest in biologically and socially important features [65]. It is believed that music has evolved from acoustic structures used by various species for emotional communication [66].

Studies on the effects of sound waves have been performed on laboratory animals. Properly selected music can relieve stress, which translates to better immune response to disease and cancer. In addition, music is a synthesis of multiple factors: rhythm, frequency, tone, loudness, sound. These aspects determine how the sound waves affect the body. For example, frequencies between 4000 and 16,000 Hz promote dopamine synthesis, and the subsequent increase in dopamine reduces blood pressure via D_2_ receptors (dopamine receptors). Baroque music, with a tempo of 60 bpm (beats per minute), significantly improves memorization and learning by activating both brain hemispheres (left and right) [19,22,67]. Given this evidence of the effect of acoustic waves, such solutions were then recommended to livestock farmers. Enriching the livestock environment with non-natural auditory sensations, such as classical music, may have highly beneficial effects on the animals’ well-being [68]. Furthermore, music can be used to facilitate specific behaviors—cows entered the milking compartments of an automatic milking system more readily when they heard familiar melodies [12]. In another study, music was shown to pacify cattle in a slaughterhouse, with the added benefit of improving working conditions for the employees. The handlers were thus more positively inclined towards their work and less likely to mistreat the animals (e.g., by abusing them) [69].

#### 3.2.1. The Effect of Sound Waves on Cattle

Cattle are exposed to various types of stress (e.g., thermal, chronic). Avoiding stress is particularly crucial for dairy cows, since milk yield maximization is a priority. High-yielding cows are taxed by metabolic and psychological processes, which leads to problems such as reduced milk production, as well as lower protein and fat content [70,71]. Noise is one of the major stress triggers. Cattle must be raised in a quiet environment, since abrupt, loud sounds may negatively impact milk yields [59,72]. Cows exposed to sounds of 80 dB feed less, become restless, and have higher heart rates (Table 1). Noise can also cause reproductive disorders—disrupting the estrus cycle, conception, and reproductive system function [59].

Cattle possess relatively sensitive hearing. Their range of audibility is between 23 Hz and 35 kHz (most sensitive to sounds of 8 kHz), with the lower limit of audibility being 21 dB [73]. In addition to the sound level, the genre of the music affecting the animal is important as well. This is particularly well-illustrated by studies on bovines, showing that cattle respond physiologically to different genres of music, ranging from classical, hard rock, to Latin American [52,74]. However, researchers sometimes fail to mention the type of music played during the study in the methodology, or the details are not readily accessible, which makes a broader analysis problematic. For example, Uetake et al. (1997) conducted an experiment on Holstein cows at mid and late lactation. Music was played for the animals as they were milked in an automatic milking system (AMS). The authors concluded that the sounds stimulated behaviors that indicate readiness for milking. Unfortunately, no data were obtained on the genre used or the sound level [12]. Kemp (2019) conducted a more in-depth experiment to investigate which music genres are favored by cows. The trials were carried out on 10 Jersey cows and Jersey x HF crosses. Each day, a different music genre was played (randomly shuffled songs) via a Bluetooth speaker set in the middle of the milking parlor. The cows were milked daily while monitoring basic parameters, milk yields (the figures have been converted from gallons to kilograms) and behavior. The results are detailed in Table 2.

No values were given for music intensity or tune, but the provided metrics give a general idea of which musical genres are the most/least stressful for cattle. The highest milk yield was recorded for the control group (no music), though closely followed by one of the musical-lyrical genres—the lullaby. The authors note that animals—in this case, cows—may have individual musical preferences, just like humans do. This is illustrated by the “behavior” column of Table 3 [15]. In general, classical music is a popular choice for cow farms, as exemplified by Crouch et al. (2019). Holstein Friesian cattle were exposed to multiple tracks, including classical music from “The Classical Chillout Gold Collection”. Compared to the control (no music) group and the group exposed to country music (John Denver’s “Legends”), the classical sound was observed to limit locomotory behaviors, vocalizations and tongue rolling. Furthermore, the herd was more likely to engage in positive social interactions. The animals proved to be more relaxed, taking more time to rest and ruminate. Similar reactions were recorded by Crouch et al. (2019) when an audiobook (“Harry Potter and the Philosophers Stone”, narrated by Stephen Fry) was played [52]. Indian instrumental music has been demonstrated to raise milk yields by as much as 12.64% [75]. On the other hand, rock music produced the opposite effect in cows. Apart from reducing milk yields, it also caused elevated levels of LDH (lactate dehydrogenase) in blood plasma and, at 90 dB, disrupted glucose metabolism and insulin secretion [74,76]. Similar effects have been observed for African Percussion Music—which employs polyrhythm, rattles, iron gongs, sticks, calabashes and marimba (in addition to LDH, this genre also prompts elevated levels of globulins and glutamic-pyruvic transaminase in blood plasma)—Latin American music, and, according to JiaJia et al. (2015), even folk music (most likely of Chinese origin, given the country of publication and the nationality of the authors) [74,77,78]. Based on these examples, it would seem that music genres with a subdued, natural sound (such as classical or relaxing music) are the best and safest choice for both cattle and humans. Fast, heavy and rhythmic music can prove counterproductive. Despite the many benefits of music, periods of no auditory stimulation are equally essential to relaxation, and should also be provided [52].

#### 3.2.2. Impact of Sound Waves on Poultry

The sense of hearing is essential to birds and is highly sensitive to the frequency range of 10–12,000 Hz [80]. In hens, the region of best sensitivity to sound ranges from 3000 to 5000 Hz [81]. The ear must adapt to process complex auditory stimuli to accurately recognize temporal and spectral information contained in the vocalizations of other similar birds [82]. Chick hearing starts to develop as early as at the embryonal stage. The 20th century consensus held that chick embryos could “hear” as early as day 10 of incubation [83], but later research failed to find auditory activity at such an early stage [84]. Studying the ganglion neurons innervating the basilar papilla of chicken embryos between 12 and 18 days of incubation, Jones et al. (2006) found that cochlear ganglion neurons showed profound insensitivity to sound between 12 and 16 days of incubation, which the authors referred to as the “prehearing” stage (Figure 4). Afterwards (starting at about day 15 of incubation, and most likely from day 16 to 18), responses to external sounds and frequency selectivity begin to emerge, with a CF (characteristic frequency range) of 170 to 4478 Hz—the embryonic cochlea detected and encoded outside sounds [85].

Interestingly, despite having less developed sound perception than mammals and no auris externa, birds still react to sounds quickly and sharply. Chicks have limited hearing at an early age, being mostly sensitive to the quiet and low clucking of the hen, whereas the latter readily picks up the high cheeping of the young [86]. The tonotopic map (from Greek *tono*—frequency; *topos*—place) of the embryonic chicken cochlea matures and becomes relatively stable 19 to 21 days after hatching [87]. Chicks first hear low-, then high-frequency sounds [88]. Noise is one of the potential stress triggers, with sudden and high-frequency noise being perceived as the most distressing [89]. Excessive stress induced by noise can lead to decreased pH in the muscles, producing low-quality meat, e.g., PSE (pale, soft, exudative) meat, as well as hyperactivity in birds—e.g., nervous wing-flicking—resulting in DPM (deep pectoral myopathy) [90,91]. Abnormal PSE meat accounts for 5 to 40% of the total poultry production, meaning that it is not only a quality issue, but a financial one—using the example of turkey meat producers, losses can be as high as 4 million dollars per annum [92]. Noise can also cause weight loss—after just 7 days of exposure to loud sounds (5 min noise, 10 min no noise), broiler chickens had 6% less weight than the control at 70 dB and 6.36% less at 80 dB [93]. Chicks exposed to specific sound stimuli at 65 dB (such as fan noise or other chicks’ vocalizations) and 90 dB (background noises, as well as motor vehicle and aircraft noise) had a higher heterophil-to-lymphocyte ratio, which may indicate acute inflammation (heterophil levels rise with stress response and the resultant release of catecholamine hormones, e.g., corticosterone). Furthermore, the chicks entered into longer periods of tonic immobility, another indicator of high stress [94,95]. High-level sustained noise can also decrease egg weight and production in hens. Noise at 80 dBA (the ‘A’ denotes that the measurements were carried out on the basis of the human perception of loudness) was found to result in an abnormal egg rate of 5.5%, which rose to 14.6% at 100 dBA [96,97]. Other observed complications include higher rates of dead embryos in eggs and genetic changes, the latter causing limb and beak deformation in subsequent generations [98]. Significant also is the time of exposure to the unpleasant auditory sensations. Animals should not be kept in spaces where the continuous noise levels exceed 85 dB, though sudden sounds should also be avoided [99,100].

So far, there has been little research on specific musical genres that would boost stress resilience in chickens, and those few studies that have been conducted sometimes produce conflicting results. Another complication is that studies often fail to cite the specific songs played during the given experiment—a limitation also present in research on cattle. Despite these difficulties, it is clear that classical music (in general—the specific pieces are not mentioned) played at 75 dB (5 h for 3 days) may increase susceptibility to stress in birds and extend tonic immobility times—unlike in cows, for which the opposite effect was noted [94]. Another study on broiler chickens has shown that A. Vivaldi’s “Four Seasons” when played at 75 dB (3 h a day from the 1 to 35 days of age) boosts live weight gain and reduces blood corticosterone up until 7 days of age. It can thus be concluded that the piece has a stress-reducing effect within the considered period—significant differences were observed only in the first week of age [101]. The last finding is contradicted by another study using the same piece (“Four Seasons” by A. Vivaldi) at 75 dB (1 h of music on, 1 h off). Combined with other factors (imprinting and environmental enrichment), classical music was shown to have a significant effect on body weight in 8-week-old chicks, which grew to a larger size than the control birds. However, no significant effect was found for feed intake and mortality [13]. Nevertheless, it seems that not all musical pieces have significant effects on chicken live weight and carcass quality—one such example is the Piano Concerto No. 2 by W.A. Mozart, played at 75 dB [102]. There have also been attempts to investigate how exposure to dinner music and rock’n’roll—both at low (75 dB) and high (85 dB) sound levels—affects meat-type chicks. Though no significant effects were observed, the results (Table 3) do reveal that high-level rock’n’roll did cause lower final weight (at the time of the slaughter, the exposed animals were approx. 4% less heavy than the control group) and carcass weight (lower by over 4% compared with the control). By contrast, dinner music caused slight increases in chick live weight (final live weights over 1% higher than the control) when played at 85 dB, and similar increases in carcass weight when played at 70 dB (carcass weight was approx. 2% higher than in the control). Furthermore, behavioral changes were only noted at initial exposure to the sound waves (e.g., moving away from the speakers, piling up in the corners of the holding area), subsuming after the first week [79].

#### 3.2.3. The Influence of Sound Waves on Pigs

The effects of music on health and behavior have also been commonly studied in pigs. These animals are known to vocalize to communicate, each vocalization having distinctive acoustical characteristics, such as (D—duration; P—main energy-resonance frequency) [14]:Isolated piglets: D—0.34 s; P—3500 Hz.Piglets processed by humans: D—0.81s; P—3700 Hz.Sows during nursing: D—0.15 s; P—1000 Hz.Sows during farrowing: D—0.1 s; P—3000 Hz.

Pigs react to sounds from 42 Hz to 40.5 kHz, with a region of best sensitivity from 250 Hz to 16 kHz [103]. High-volume sounds that meet the definition of noise lead to aggressive behavior and weakened immunity in pigs [104]. Researchers studying the effect of music on pigs have been particularly keen on piglets as a subject of study, with a large body of research having been published on the topic [105,106,107]. Removal from the sow is a critical and highly stress-inducing moment in rearing piglets. Piglets exposed to music pre-weaning developed a conditioned response. Piglets that then listened to music post-weaning were quicker to return to relaxed play behavior, which translates to improved well-being and lower number of injuries [107]. Despite the lack of success in obtaining information on the music genres used in past research, it is clear that researchers favor classical music—primarily compositions by Mozart and Vivaldi, i.e., sounds of piano and violin accompanied by a symphony orchestra. For example, Sonata for Two Pianos in D (K.448) by W. A. Mozart, played at 60–70 dB stimulates behavioral activity in pigs—promoting tail movement, exploration and play, though these behaviors diminished with time. Reduced cortisol secretion was also observed during the 8-day music treatments, interpreted as a sign of low stress. Furthermore, pigs exposed to Mozart for 60 days had a stronger immune response, with higher levels of IgG (immunoglobulin G), IL-2 (interleukin-2), IFN-γ (interferon gamma), and lower levels of IL-4 (interleuikn-4) [104]. Another example is “Four Seasons” by A. Vivaldi, already tested in research on fowl [93]. In pregnant sows, the piece (played at 71.13 dB) induced deeper relaxation, as evidenced by metrics such as lower respiration rate, less stereotypies and better interactions with humans [108]. In contrast, piglets did not take well to the piece by the famous violinist—the music proved distressing to the animals and, thus, disrupted their rest [106]. Although the adverse reaction to classical music might seem odd, the negative physiological effects of rock’n’roll on different animal species are indisputable [102]. The same genre played to pigs at 80–85 dB (twice daily during feeding) had a negative effect on the daily growth rate (DGR) and feed conversion ratio (FCR). The treated specimens fed less regularly and had significantly lower DGR than the other groups (silence and light music) throughout the experiment (Figure 5). Rock’n’roll caused pigs to grow slower and was, in this respect, similar to exposure to 120 dB noise (though it is as detrimental even at lower levels of 80–89 dB [109]).

The authors have thus rightly concluded that the intensity of the sound waves is not the only significant variable—frequency and rhythm play a part as well [56,104]. This has been partially corroborated by another study, which also examined musical preferences in piglets [105]. The specimens were divided into five experimental groups—a no-music control and:String–slow (SS; 65 bpm);String–fast (SF; 200 bpm);Wind–slow (WS; 65 bpm);Wind–fast (WF; 200 bpm).

All piglets were more likely to choose chambers where SS and WF music was played, and showed behavioral differences compared to the control: SS music increased lying time, and WF music promoted exploration, with piglets intermittently walking and lying (same as the WS group). SF music encouraged tail-wagging [105].

## 4. Conclusions

This review of the available literature demonstrates that music can be used to enrich the living environment of livestock. With the right choice of music genre, sound level and tempo, music can alleviate the adverse effects of noise and, thus, reduce stress. It should be kept in mind that silence is equally important and necessary for the well-being of the animals, and that playing loud music to animals—such as rock’n’roll or heavy metal—should be avoided, as it can negatively affect their health. The use of properly selected music in intensive livestock production improves welfare. Intensively housed livestock have to deal with more stress triggers, which weakens their tolerance to stress-inducing factors and, thus, leads to compounding health problems. One interesting line of inquiry would be to create a set of specific musical items and tracks composed specifically for a given species of livestock, tailoring the sound of the instruments, the tune of the song and the sound frequency appropriately.

## Figures and Tables

**Figure 1 animals-11-03572-f001:**
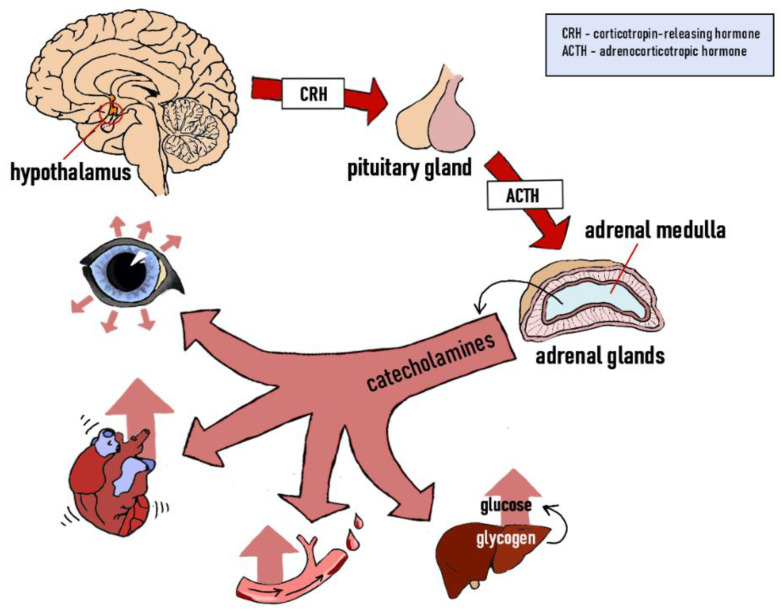
Stress response—the sympathetic nervous system (SNS). Own elaboration based on [36].

**Figure 2 animals-11-03572-f002:**
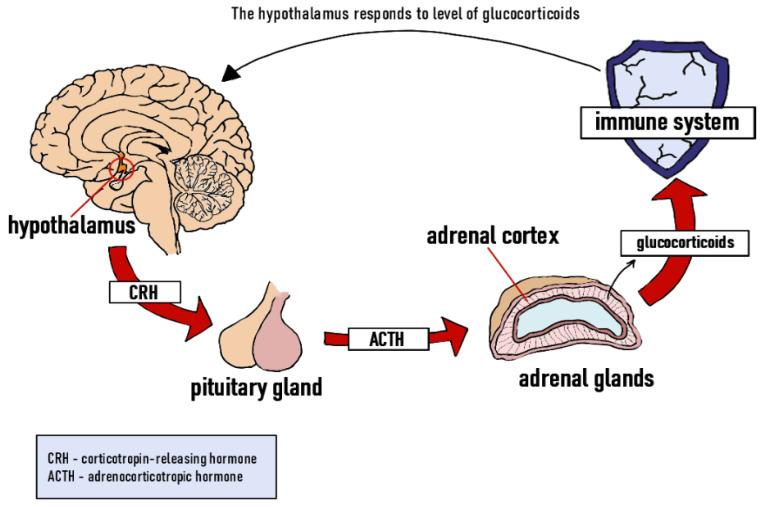
Stress response—the hypothalamic–pituitary–adrenal axis (HPA). Own elaboration based on [36,40,41,42].

**Figure 3 animals-11-03572-f003:**
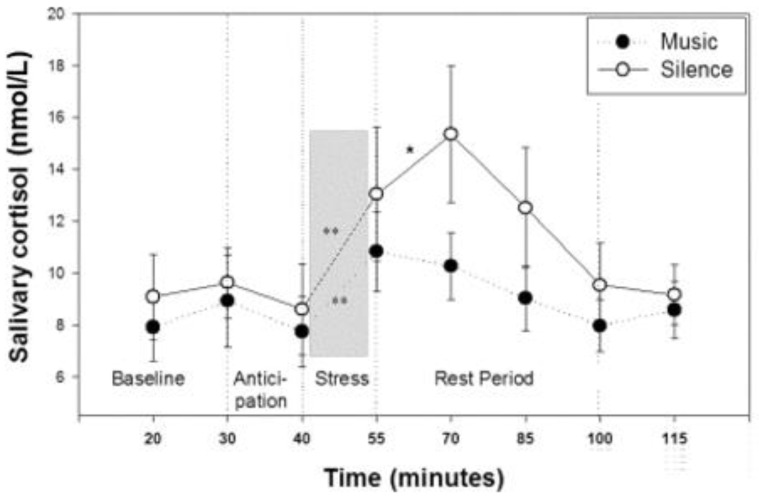
Means and standard errors of salivary cortisol concentration from 20 to 115 min after subjects’ arrival; ** *p* < 0.01; * *p* < 0.05 [57].

**Figure 4 animals-11-03572-f004:**
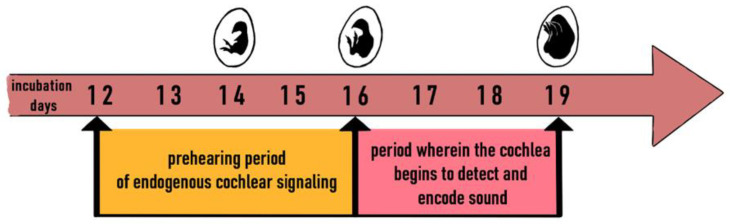
The perception of sounds by the chicken embryo from 12th to 19th day of incubation. Own elaboration based on [85].

**Figure 5 animals-11-03572-f005:**
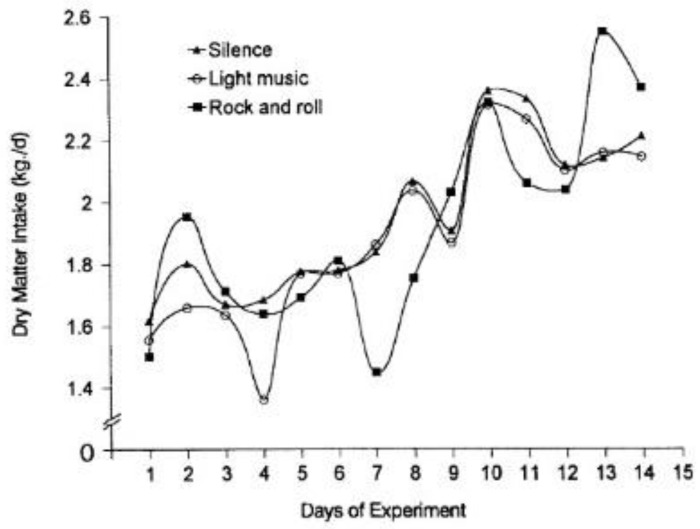
Daily dry matter intake of the pigs in the treatments [56].

**Table 1 animals-11-03572-t001:** Impact of noise of varying intensity on cattle. Own elaboration based on [59].

Noise Volume [dB]	The Effects of Noise
80 dB	excessive anxiety, increased heart rate, reduction in feed intake
90–95 dB	anxiety, frequent bowel movements, muscle tension, increased heart rate, reduction in rumen contractions, food retention
≥100 dB	morphological and biochemical changes in blood (increase in blood glucose levels, development of leukocytosis)

**Table 2 animals-11-03572-t002:** Influence of the music genre on basic life parameters, milk yield and cow behavior (N = 10). Own elaboration based on [15].

Day	Music Genre	Milk Yield [kg]	HR ^1^ [bpm]	Respiratory Frequency [bpm]	Behavior
1	Control group	51.58	65	22	8 cows—relaxed behavior ^2^, 2 cows—slightly alert ^3^
2	Country	46.37	60	18	9 cows—relaxed behavior, 1 cow—slightly alert
3	Rock	45.9	66	16	3 cows—relaxed behavior, 7 cows—alert
4	Jazz	44.48	57	16	7 cows—relaxed behavior, 3 cows—slightly alert
5	Reggae	45.42	61	18	10 cows—relaxed behavior
6 ^5^	Pop	38.8	63	19	6 cows—slightly alert, 4 cows—relaxed behavior
7 ^5^	Classical music	43.5	59	14	10 cows—relaxed behavior (5 ofwhich were ruminating)
8	Opera	34.54	63	15	9 cows—relaxed behavior, 1 cow—slightly alert
9	Rap	42.11	59	18	3 cows—relaxed behavior, 3 cows—slightly alert, 4 cows—confused ^4^
10	Hip Hop	45.42	64	20	7 cows—relaxed behavior, 3 cows—slightly alert
11	Lullaby	48.26	56	15	10 cows—relaxed behavior (1 cow fell asleep)
12	Heavy Metal	46.84	67	21	1 cow—relaxed behavior, 9 cows—alert

^1^ HR—heart rate.^2^ relaxed behavior—vitals normal or below normal, lowered head, lowered eyelids, rumination, grooming. ^3^ slightly alert/alert—eyes wide open, head raised. ^4^ confused—wandering eyes and ears, head raised. ^5^ 6, 7—the result of the daily milk yield may be distorted due to an incident (beef calf intrusion into the barn and possible milk intake from dairy cows).

**Table 3 animals-11-03572-t003:** Treatment means for live weight, warm dressed carcass weight and percent yield [79].

Treatment	Live Weight (g)	Carcass Weight (g)	Yield (%)
1—control	2020	1461	72.3
2—low level dinner music	2045	1491	72.9
3—high level dinner music	2053	1480	72.0
4—low level rock and roll music	2032	1486	73.1
5—high level rock and roll music	1942	1398	72.0

## Data Availability

Not applicable.

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
