# Peer review of "The Effect of Music on Livestock: Cattle, Poultry and Pigs"

_animals, 2021, doi:10.3390/ani11123572_

Round 1

Reviewer 1 Report

This is a very well written review. My comments/suggestions are minor, but would be helpful for the reader:

L. 42. please include: perceived. "...discerning about the perceived quality and..."

L139. I believe the finding of Rushen. 1999. Fear of people by cows and effect on milk yield, behavior, and heart rate at milking. J. Dairy Sci. 82:720-727 or Hemsworth et al. Relationships between human-animal interactions and productivity of commercial dairy cows. J. Anim. Sci. 2000. 78:2821–2831 might fit better than "heat stress" at this point 

Table 2: The Kemp study is cute but a bit superficial. Please include (N=10) in the title and also add a footnote for day 6 & 7 that the lower milk yield might also be due to a beef calf that broke in the milk cow barn at night. Also, for readability I'd add a column for the day.

Table 3: Switch 2 and 3 to be consistent from low to high within the table

Figure 4: It took me a while to understand what E12 meant. For readability I would remove the "E" and just place a "incubation days" on/in the left side of the arrow

References. Please review them for completeness, e.g. #36, 41,48, 70, 74 etc. are incomplete

Author Response

Dear Reviewer 1

Authors are very grateful for valuable comments to improve our paper. We have implemented all the remarks indicated and believe the paper is now suitable for publication. We have introduced changes in the text using the “track changes” tool in the Word program. To support visibility of the modifications made, we were have added numbers of lines where they were implemented and marked them in yellow.

The following remarks have been addressed in the manuscript:

  1. 42. please include: perceived. "...discerning about the perceived quality and..."

Line 42: Added

  1. I believe the finding of Rushen. 1999. Fear of people by cows and effect on milk yield, behavior, and heart rate at milking. J. Dairy Sci. 82:720-727 or Hemsworth et al. Relationships between human-animal interactions and productivity of commercial dairy cows. J. Anim. Sci. 2000. 78:2821–2831 might fit better than “heat stress” at this point

Line 150-153: Text modified slightly and a reference to the article by Rushen et al., 1999 added.

  1. Table 2: The Kemp study is cute but a bit superficial. Please include (N=10) in the title and also add a footnote for day 6 & 7 that the lower milk yield might also be due to a beef calf that broke in the milk cow barn at night. Also, for readability I'd add a column for the day.

Line 290-295, Table 2: Added N = 10 in the title of the table; Line 257 - Annotation was added to days 6 and 7 where an incident occurred prior to milk yield measurements; Added a column for the day

  1. Table 3: Switch 2 and 3 to be consistent from low to high within the table

Line 403-404, Table 3: Corrected

  1. Figure 4: It took me a while to understand what E12 meant. For readability I would remove the "E" and just place a "incubation days" on/in the left side of the arrow

Line 338-341, Figure 4: Corrected

  1. Please review them for completeness, e.g. #36, 41,48, 70, 74 etc. are incomplete

The bibliography was corrected in the position: 15, 36, 41, 48, 59, 64, 70, 74, 76.

Including language revision (please see the attached certificate). 

Reviewer 2 Report

This is an interesting study in some ways , but the authors MUST further examine the ethical issues involved with their statements considering how music could increase: yield/growth etc in intensive husbandry systems on the one hand, and then using humans physiological responses to music as a model for their reviews... this does not back "human exceptionalism", in which case humans  then can also be raised in intensive systems  where growth/ yield etc/genetic manipulation etc is maximized with music? One way around this would be to examine whether or not intensive livestock systems, with or without music or silence, is acceptable at all, for individual wellbeing of the animals or the well-being of the planet,  and at least least to point out that it may not be in which case the desired aims might be mis-laced. "tweaking" such as the right type of music might be slightly helpful for the welfare of the individuals today in intensive animal systems but we need to examine if they should continue... they are one of the major causes of reduced species diversity and climate warming. 

Some further difficulties here:

line 86. Selye's Stress response is correctly understood, but it is not "stress" itself that is generally the problem in intensive husbandry of cattle, poultry and pigs, it is more generally prolonged stress  ( distress ) which changes the hormonal and other mental and body responses of the individual. This is called "distress" and has been widely used in the welfare literature as distinct from "stress".

line 125. There are many more triggers of "stress" ( loud noises, sudden light changes etc etc etc), and "distress". Human contact is NOT necessarily one of these, it depends entirely, like the music on the KIND of human contact (e.g. milk yields decline with rough handling, or even handling by humans who are themselves "stressed").   Separating humans from the animals may actually  reduce welfare rather than increase it! It depends on the humans' behaviour as does the type of music.( line 139) 

176 You use terms such as "relaxing music" with no definition until later, suggest you define your terms first then review the effects. Does this mean slow, or fast, loud or soft ( soft beats or even squeals can be very distressing), what type of harmonies , some are much more acceptable  than others, pitch and the overtones etc etc.  

207 music does not always reduce stress, it can increase it as you say yourselves. Rewrite 

Silence part is interesting, but what about noises off such as crashing of gates, moving of individuals in the group, groans and grunts etc etc, how did you ensure non of these? or did you?? In which case needs reconsidering. 

Table 2 "chewing~" do you mean cuddiing or ruminating if so this is usually associate with relaxation, and you need to mention this, there are EEG's etc references on this 

426 use of sound waves to increase welfare... but not sound waves, just the type! as you have shown. 

I think there is a need of rewriting parts of this paper and clarifying and particularly at least mentioning both the animal welfare and the environmental welfare concerns of intensive husbandry rather than just tweaking because there is an idea here using humans as a model, does threaten human exceptionism, so may be we need to recognize this and get rid of intensive systems, although some sorts of music could enrich animals worlds like it does ours.  

Author Response

Authors are very grateful for valuable comments to improve our paper. We have implemented all the remarks indicated and believe the paper is now suitable for publication. We have introduced changes in the text using the “track changes” tool in the Word program. To support visibility of the modifications made, we were have added numbers of lines where they were implemented and marked them in yellow.

The following remarks have been addressed in the manuscript:

  1. line 86. Selye's Stress response is correctly understood, but it is not "stress" itself that is generally the problem in intensive husbandry of cattle, poultry and pigs, it is more generally prolonged stress ( distress ) which changes the hormonal and other mental and body responses of the individual. This is called "distress" and has been widely used in the welfare literature as distinct from "stress".

Line 98-110: changed "stress" to "distress"

  1. line 125. There are many more triggers of "stress" ( loud noises, sudden light changes etc etc etc), and "distress". Human contact is NOT necessarily one of these, it depends entirely, like the music on the KIND of human contact (e.g. milk yields decline with rough handling, or even handling by humans who are themselves "stressed"). Separating humans from the animals may actually  reduce welfare rather than increase it! It depends on the humans' behaviour as does the type of music.( line 139)

Line 145: The phrase  “human contact” has been changed into “man-animal interaction”.

Line 150-153: a fragment has been added that improper care adversely affects animal body

  1. 176 You use terms such as "relaxing music" with no definition until later, suggest you define your terms first then review the effects. Does this mean slow, or fast, loud or soft ( soft beats or even squeals can be very distressing), what type of harmonies , some are much more acceptable than others, pitch and the overtones etc etc. 

Line 184-194: The notion of relaxing music has been explained

  1. 207 music does not always reduce stress, it can increase it as you say yourselves. Rewrite

Line 243: It has been corrected to “Properly selected music can”

  1. Silence part is interesting, but what about noises off such as crashing of gates, moving of individuals in the group, groans and grunts etc etc, how did you ensure non of these? or did you?? In which case needs reconsidering.

Line 205-218: A fragment about silence has been added to the revised version

  1. Table 2 "chewing~" do you mean cuddiing or ruminating if so this is usually associate with relaxation, and you need to mention this, there are EEG's etc references on this

Line 278, Table 2: It has been corrected to “ruminating”

  1. 426 use of sound waves to increase welfare... but not sound waves, just the type! as you have shown.

Line: 466-467: It has been corrected to “Use of properly selected music”.

All remarks of Reviewers have been considered and implemented in the revised manuscript, including language revision (please see the attached certificate). The only remark we do not understand is:

„I think there is a need of rewriting parts of this paper and clarifying and particularly at least mentioning both the animal welfare and the environmental welfare concerns of intensive husbandry rather than just tweaking because there is an idea here using humans as a model, does threaten human exceptionism, so may be we need to recognize this and get rid of intensive systems, although some sorts of music could enrich animals worlds like it does ours”. 

which implies that it is mainly the intensive production system that can be claimed responsible for diminished animal welfare. In our opinion, it is only somewhat true, because birds are exposed to various stress factors also in the free-range production system. Our manuscript does not address a man as a model – the stressor, but the use of sounds in various rearing systems. Because consumers have the worst connotations with animal rearing in the intensive production system, scientists usually undertake studies under this system’s conditions and, hence, the highest number of research works address this production system. However, sounds ought to be deployed in every production systems, because in each of them animals are exposed to stress stimuli.

Yours Sincerely,

Monika Michalczuk
